# Assessing Automated Facial Action Unit Detection Systems for Analyzing Cross-Domain Facial Expression Databases

**DOI:** 10.3390/s21124222

**Published:** 2021-06-20

**Authors:** Shushi Namba, Wataru Sato, Masaki Osumi, Koh Shimokawa

**Affiliations:** 1Psychological Process Team, BZP, Robotics Project, RIKEN, 2-2-2 Hikaridai, Seika-cho, Soraku-gun, Kyoto 6190288, Japan; 2KOHINATA Limited Liability Company, 2-7-3, Tateba, Naniwa-ku, Osaka 5560020, Japan; osumi@kohinet.com (M.O.); shimokawa@kohinet.com (K.S.)

**Keywords:** action unit, automatic facial detection, facial expressions, machine analysis, sensing dynamic face

## Abstract

In the field of affective computing, achieving accurate automatic detection of facial movements is an important issue, and great progress has already been made. However, a systematic evaluation of systems that now have access to the dynamic facial database remains an unmet need. This study compared the performance of three systems (FaceReader, OpenFace, AFARtoolbox) that detect each facial movement corresponding to an action unit (AU) derived from the Facial Action Coding System. All machines could detect the presence of AUs from the dynamic facial database at a level above chance. Moreover, OpenFace and AFAR provided higher area under the receiver operating characteristic curve values compared to FaceReader. In addition, several confusion biases of facial components (e.g., AU12 and AU14) were observed to be related to each automated AU detection system and the static mode was superior to dynamic mode for analyzing the posed facial database. These findings demonstrate the features of prediction patterns for each system and provide guidance for research on facial expressions.

## 1. Introduction

Facial expressions can facilitate social interaction by conveying diverse messages including thoughts, perceptions, emotions, plans, and actions [1]. To describe all of the movements involved in facial expressions, Ekman and his colleagues developed the Facial Action Coding System (FACS) [2]. FACS is the most objective and comprehensive system to describe facial movements from holistic facial expressions without any inference. FACS can describe all facial movements by combining facial components, termed action units (AUs), based on anatomy. For example, a movement of the zygomatic major muscle to pull the corners of the lips can be described as AU12, and contraction of the outer orbicularis oculi muscle, which raises the cheeks, can be termed AU6. Table 1 presents a list of AUs used in the current study. Manual FACS coding has shed light on numerous psychological experiments related to facial activity [3,4].

Because manual FACS coding has the disadvantage of being time consuming, automatic detection of FACS AUs has been an active area of research [5]. Automated facial AU detection systems are available as both commercial tools (e.g., Affectiva, FaceReader) and open-source tools (e.g., OpenFace, [6,7] and Automated Facial Affect Recognition (AFAR) [8,9]). One study found that OpenFace and AFAR generally performed similarly, but average results were slightly better for AFAR [5]. In addition, the performance of automatic AU detection by the current commercial systems has been checked in many ways. Manual FACS coding and FaceReader were compared [10] and applied to static posed facial images and revealed medium scores for the balance of precision and recall (i.e., F1 scores: WSEFEP = 0.67; ADFES = 0.66). Moreover, [11] showed that FaceReader had a moderate F1 score (0.63) using private facial stimuli that had been composed by mimicry of prototypical expressions or deliberate enactment of emotions. Although the systems are not perfect, previous studies have supported the validity of automated AU detection systems to a certain extent.

However, three issues with automatic AU detection systems remain. First, system accuracy when targeting one facial stimulus does not guarantee accuracy when targeting another facial stimulus, i.e., their generalizability across various domains is unproven. The interest in affective studies has shifted from posed and static facial images to more spontaneous and dynamic facial movements [12,13,14,15]. In this situation, the most important question for affective computing systems is “to what extent does the automatic AU detection system generalize to other data in the various domains.” To the best of our knowledge, how well open-source AU detection systems and even commercial AU detection software can perform with spontaneous and dynamic facial movements compared to human FACS coding remains unknown. In addition, the extent of difficulty estimating each facial movement on FACS also depends on the context in which the facial expressions occur; [16] found that automatic emotion recognition performance was generally better for posed than for spontaneous expressions. Moreover, it is predicted that even with automated AU detection systems, accuracy will be higher with posed rather than with spontaneous facial expressions.

Second, there has been no systematic comparison of AU detection accuracy among systems. FaceReader is a commercial software designed to analyze facial expressions, whereas OpenFace [6,7] is the dominant shareware automatic facial computing system for many applied situations [17,18], and AFAR is an open-source, state-of-the-art, algorithm-based user-friendly tool for automated AU detection [8,9]. Although comparisons of the performance of these systems are interesting for newcomers and important in terms of system selection, to our best knowledge, no studies have compared these three tools as of yet.

Finally, the systems’ AU detection biases remain to be explored. It is recognized that the automated facial action detection systems sometimes cause confusion during each AU estimation and [19] reported that the OpenFace algorithm might confuse AU10 (upper lip raiser) with AU12 (lip corner puller). A systematic evaluation of automated machines’ confusion biases among the AUs would be useful, as it would provide researchers with an insight into which facial movements are targeted and which confusions should be avoided.

Here, we addressed these issues by comparing the performance of three AU detection systems: FaceReader, OpenFace [6,7], and AFAR [8,9]. We used a cross-domain database such as in-the-wild data from YouTube videos, posed facial movements expressing emotional states and social facial expressions during conversations. Although F1 scores are the most common metric for evaluating automated AU detection systems [20], this study employed the area under the receiver operating characteristic curve (AUC). Jeni, Cohn, and De La Torre showed that AUC provides more robust results than other indices (e.g., F1 and Cohen’s kappa) when using imbalanced data where the number of positive examples is often small, such as a spontaneous facial expression database [21]. Thus, this paper only focused on AUC values, which can be considered a robust index for avoiding redundancy. Other comprehensive information such as the F1 score, negative agreements, and others are described in the Appendix A (https://osf.io/vuerc/?view_only=8e9077dbeaec424bad5dde5f07555677) for the sake of transparency of the results. Moving beyond previous research [7,8,9,10,11], this study aimed to provide bias scores using AUs’ co-occurrence and predictions. We hypothesized that all systems would show above-random performance (i.e., AUC = 0.50). The current study also hypothesized that all machines would show better AU detection performance with posed rather than with spontaneous facial expressions. Although there might be little difference in performance between OpenFace and AFAR [5], we had no directional hypothesis for the commercial software. As for the machines’ bias scores, we evaluated each system systematically and exploratively.

In automated facial action detection systems, there are several options for calibration, feature extraction, and various pre-trained models. For example, OpenFace typically uses the pre-trained convolutional experts constrained local model (CE-CLM) [22] to obtain facial landmarks, but it can also use the constrained local neural field model (CLNF) [23]. To provide guidance for users, we further investigated these options.

## 2. Materials and Methods

### 2.1. Facial Datasets

All of the following facial databases were manually coded frame-by-frame by FACS coders. In addition, we used the threshold A-intensity as the occurrence of AUs. To provide information on databases, all frequencies of manually coded AUs are described in Appendix A (https://osf.io/nwpt2/?view_only=8e9077dbeaec424bad5dde5f07555677).

#### 2.1.1. In-the-Wild SNS Data

Aff-wild2 is an in-the-wild database with annotations for AUs [24,25,26,27,28,29]. These data were collected using YouTube. The current study used 30 clips (50% female), all part of training videos, not containing two or more people in a single file. The names of the videos used are provided in the Appendix A (https://osf.io/bxj4r/?view_only=8e9077dbeaec424bad5dde5f07555677). The total number of video frames was 169,686. This database contained the following AU annotations: AU1, AU2, AU4, AU6, AU12, AU15, AU20, and AU25.

#### 2.1.2. Data from Posed Expression of Emotions

The extended Denver Intensity of Spontaneous Facial Action (DISFA+) [30] included posed facial expression data from nine young adults (56% female). While this dataset had several posed facial expressions in response to instructions to express particular emotions, the current study used only displays from individuals who were instructed to express emotional words (anger, disgust, fear, happiness, sadness, and surprise). All videos were 12,394 frames. The coded AUs were AU1, AU2, AU4, AU5, AU6, AU9, AU12, AU15, AU17, AU20, AU25, and AU26.

#### 2.1.3. Conversation Data

The Sayette Group Formation Task (GFT) database has been used for conversational data [31]. This dataset includes unscripted social interactions within groups of three unacquainted adults. Thirty-two groups comprising a total of 96 participants (42% female) resulted in 172,800 frames, although the validated number was 133,194 frames, as the camera occasionally failed to capture facial expressions. The manually coded AUs were the following: AU1, AU2, AU4, AU5, AU6, AU7, AU9, AU10, AU11, AU12, AU14, AU15, AU17, AU18, AU19, AU22, AU23, AU24, and AU28.

### 2.2. Automatic Facial Detection Systems

The pipelines of all systems are described in Figure 1. The details of each system are provided in the following sections:

#### 2.2.1. Face Reader

The current study applied version 7 of FaceReader (Noldus, 2016). According to the documentation, facial landmarks derived from the Active Appearance Model (AAM) are used to perform the neural network model. In parallel, facial shape images directly obtained from the facial data are used to perform a deep artificial neural model. The final output is the result of integrating the two models. However, as [7] indicated, there is no transparency regarding the algorithms and training data. The AU module can provide coding for 20 FACS AUs (1, 2, 4, 5, 6, 7, 9, 10, 12, 14, 15, 17, 18, 20, 23, 24, 25, 26, 27, and 43).

#### 2.2.2. OpenFace

This system, which was developed by [7], has pre-trained models for the detection of facial landmarks. The default pre-trained model is the CE-CLM, which trained varying pose and illumination data and in-the-wild data. The other model is the CLNF model. For the main analysis, we used the default CE-CLM model to detect the facial landmark. OpenFace 2.0 also uses dimensionality-reduced histograms of oriented gradients (HOGs) and facial shape images from the CE-CLM model as features. To recognize AUs, they used a linear kernel support vector machine. Further information is provided in [7]. It should be noted that this AU detection model was trained using many facial databases (e.g., SEMAINE, [32], BP4D, [33], Bosphorus, [34], DISFA, [35], FERA 2011, [36], and UNBC-McMaster shoulder pain expression archive database, [37]). The 18 recognizable AUs include 1, 2, 4, 5, 6, 7, 9, 10, 12, 14, 15, 17, 20, 23, 25, 26, 28, and 45.

#### 2.2.3. AFAR

The Automated Facial Affect Recognition system (AFAR) used Zface [38], which accomplishes 3D registration from 2D videos, resulting in normalized face videos composed of 200 × 200-pixel images of faces with an 80-pixel interocular distance. After image registration, methods to extract features can be applied, including histograms of oriented gradients (HOGs), scale-invariant feature transform (SIFT) and raw texture. For the main analysis, we used HOGs. Using these input data, a three-layered convolutional neural network (CNN) architecture trained the probabilities of AU occurrences. This system was only trained by Extended BP4D+ [39]; more information is reported in [8,9]. The target AUs are the following: 1, 2, 4, 6, 7, 10, 12, 14, 15, 17, 23, and 24.

### 2.3. Evaluation Metrics

To assess the performance of automated facial action detection systems, we focused on a receiver operating characteristic curve and the corresponding area under the curve (ROC-AUC), which can be considered a robust index for imbalanced data such as AUs derived from facial databases [21]. Thus, we calculated the AUC scores using manual AU-coded data and continuously predicted AU scores. In keeping with open science practices for transparency of results, other indices such as F1 scores, detection rate, sensitivity, prevalence, negative agreement, and others are available in the online Appendix A (https://osf.io/vuerc/?view_only=8e9077dbeaec424bad5dde5f07555677).

Moreover, we calculated the correlation matrix between manually coded AUs and the AUs predicted by each system. Then the difference between the system-predicted AU co-occurrence and the actual AU co-occurrence was used to obtain the AU detection bias of systems, computed by the following formula:
(1)$$AU\;bias\;scores=\frac{\Sigma \left(AUx_{predicted}-\bar{AU}x_{predicted}\right)*\left(AUy_{predicted}-\bar{AU}y_{predicted}\right)}{\sqrt{\Sigma {\left(AUx_{predicted}-\bar{AU}x_{predicted}\right)}^{2}*\Sigma {\left(AUy_{predicted}-\bar{AU}y_{predicted}\right)}^{2}\;}}\;-\;\frac{\Sigma \left(AUx_{observed}-\bar{AU}x_{observed}\right)*\left(AUy_{observed}-\bar{AU}y_{observed}\right)}{\sqrt{\Sigma {\left(AUx_{observed}-\bar{AU}x_{observed}\right)}^{2}*\;\Sigma {\left(AUy_{observed}-\bar{AU}y_{observed}\right)}^{2}}}$$
where $AUx_{*}$ is the *x*th AU that the system has predicted or actually observed, and $AUy_{*}$ is the *y*th AU that the system has predicted or actually observed. If this value is close to 0, this indicates that the system reveals the same AU co-occurrence as the actual observational data. On the other hand, if the absolute value is close to 1, this indicates that the machine was biased or overlooked the AU co-occurrence. Positive values indicate that biases have resulted in more AU co-occurrences than the actual relationships, and negative values indicate that the biases resulted in failure to capture co-occurrence relationships in the actual facial movements. The correlation matrix showing the co-occurrence of AUs for all three databases is shown in the Appendix A; https://osf.io/tw7pg/?show=view accessed on 19 June 2021).

All analyses were performed using R statistical software, version 4.0.3 (https://www.r-project.org/ accessed on 19 June 2021), alongside the ‘pROC,’ ‘data.table,’ and ‘tidyverse’ packages [40,41,42].

## 3. Results

### 3.1. Aff-Wild2 (in-the-Wild)

First, we investigated three automatic facial detection systems using in-the-wild data. To explore the machine’s ability to discriminate between the presence and absence of AUs, ROC curves were plotted, and the AUC was calculated (Figure 2 and Figure 3). The average AUC values for the three systems were as follows: OpenFace (*Mean* = 0.72, *SD* = 0.11), FaceReader (*Mean* = 0.67, *SD* = 0.08), AFAR (*Mean* = 0.65, *SD* = 0.10). Visual inspection of Figure 2 and Figure 3 indicated that all systems only showed good results, i.e., >0.80, for AU 12 (lip corner puller).

To confirm whether the system could detect AUs beyond a random level, we used one-sample *t*-tests (*mu* = 50). A significance level of *α* = 0.05/3 (0.05 divided by the number of machines) was used for these tests. All machines were able to detect the presence of AUs at a level above chance, *t*s > 1199.63, *p*s < 0.001.

We explored differences among the systems using one-way between-subjects ANOVA with system as a factor (FaceReader, OpenFace, and AFAR) for performance on every AU detection. No significant effects of the system were found, *F*(2, 19) = 0.97, *p* = 0.40, *ηG2* = 0.09.

### 3.2. DISFA+ (Posed)

Figure 2 and Figure 4 show the ROC curves and AUC scores for posed facial expressions. The average AUC scores for each system were as follows: OpenFace (*Mean* = 0.81, *SD* = 0.10), FaceReader (*Mean* = 0.55, *SD* = 0.05), AFAR (*Mean* = 0.79, *SD* = 0.08). Visual inspection of Figure 4 indicates that the AUC of AU 6 (cheek raiser) was high for OpenFace and AFAR. The AUC for AU25 was also high for OpenFace in the case of posed facial expressions. On the other hand, AU5 (upper lid raiser) was lower than other AUs.

The one-sample *t*-tests showed that all machines could detect the presence of AUs more accurately than chance, *t*s > 1556.55, *p*s < 0.001.

One-way ANOVA for AUC scores showed a significant effect of system, *F*(2, 28) = 40.76, *p* < 0.001, *ηG2* = 0.74. Multiple comparisons using Shaffer’s modified sequentially rejective Bonferroni procedure indicated that the AUC value for OpenFace was higher than that for FaceReader, *t*(24) = 8.36, *p* < 0.001, *Hedge’s g* [95%CI] = 3.41 [2.16, 4.66]. The value for AFAR was also higher than that for FaceReader, *t*(19) = 6.68, *p* < 0.001, *Hedge’s g* [95%CI] = 3.18 [1.80, 4.55].

Furthermore, to provide information for affective computing, we compared the AUC values obtained by tools after grouping videos by emotions in the DISFA+ posed facial database. Since it may be beyond the scope of this paper, we have not shown these results. More detailed results have been made available online as Appendix A (https://osf.io/vuerc/?view_only=8e9077dbeaec424bad5dde5f07555677).

### 3.3. GFT (Conversational)

With respect to social conversational data, Figure 5 shows the ROC curves and AUC scores for social conversational data. The average AUC values for each system (Figure 2) were as follows: OpenFace (*Mean* = 0.72, *SD* = 0.11), FaceReader (*Mean* = 0.58, *SD* = 0.12), and AFAR (*Mean* = 0.72, *SD* = 0.10). As the results for Aff-wild2 showed, visual inspection of Figure 2 and Figure 5 reveals that all systems only showed good results (>0.80) for AU12 (lip corner puller). However, the AUC values for lower face parts other than AU10 and 12 were relatively lower (e.g., AU14, dimpler; AU23, lip tightener). As for machine differences, AFAR and OpenFace systems differed depending on the type of AU but were similar in their performance overall. One-sample *t*-tests showed that all machines could detect the presence of AUs more accurately than chance, *t*s > 1553.82, *p*s < 0.001.

One-way ANOVA for AUC scores showed a significant effect of system, *F*(2, 36) = 5.20, *p* = 0.01, *ηG2* = 0.22. The performance of FaceReader was poorer than that of AFAR, *t*(28) = 2.78, *p* = 0.03, *Hedge’s g* [95%CI] = 1.09 [0.27, 1.92] and OpenFace, *t*(28) = 2.74, *p* = 0.026, *Hedge’s g* [95%CI] = 1.06 [0.25, 1.81].

Finally, to check whether each system performed better in a particular context, we conducted a between-subjects ANOVA with a factor of database (Aff-wild2, DISFA+, GFT) using AUC scores of each AU separately for each system. Although both OpenFace and AFAR showed no significant effect of database (*F*s < 2.73, *p*s > 0.08, *ηG2*s < 0.15), FaceReader was significantly affected by the databases, *F*(2, 31) = 4.49, *p* = 0.02, *ηG2* = 0.23. Multiple comparisons for FaceReader indicated that the AUC value for Aff-wild2 was higher than that for DISFA+, *t*(20) = 2.91, *p* = 0.02, *Hedge’s g* [95%CI] = 1.33 [0.34, 2.31].

### 3.4. Bias Evaluation

Figure 6 shows the bias matrix data for the three automated facial AU detection systems. The threshold for meaningful coefficients was set to values above.40 [11,43]. A common result for all three systems predicted that AU1 and AU2 would co-occur (*r*s > 0.45). For the two open-source systems (i.e., OpenFace, AFAR), AU14 co-occurred with other predicted AUs such as AU6, AU10, and AU12 (*r*s > 0.43). OpenFace also had a bias predicting that AU25 would co-occur with AU6 or AU12 (*r*s > 0.42) and not with AU26 (*r* = −0.49), whereas FaceReader overlooked the co-occurrence of AU6 and AU7, AU7 and AU10 or AU12, and AU10 and AU12 (*r*s < −0.51). In addition, we found that FaceReader overpredicted co-occurrence between AU7 and AU24 and between AU12 and AU25 (*r*s = 0.49). AFAR might be considered to have relatively less bias than the other two systems.

### 3.5. Within-System Comparison

As Figure 1 indicates, the pipelines of both OpenFace and AFAR have several options for automated facial movement detection. OpenFace uses the CE-CLM to detect facial landmarks, but analysis using the CLNF model is also available. AFAR uses HOGs by default, but SIFT can also be applied. In addition, FaceReader, OpenFace and AFAR can apply calibration and mean standardization manipulations to all frames for video analysis (dynamic mode). We compared the options among the systems; the results are shown in Table 2, Table 3 and Table 4.

With OpenFace, within-subjects ANOVA revealed differences by option for the posed data, *F*(3, 33) = 5.77, *p* = 0.03, *ηG2* = 0.15, and in-the-wild data, *F*(3, 21) = 3.26, *p* = 0.04, *ηG2* = 0.04. Although multiple comparisons using Shaffer’s modified sequentially rejective Bonferroni procedure did not reveal any significant differences, *t*s < 2.04, *p*s > 0.28 the default setting (CE-CLM + dynamic mode) performed best for in-the-wild data, and the static mode setting performed best for posed data (Table 2). The ANOVA analysis for conversational data was not significant, *F*(3, 36) = 0.54, *p* = 0.66, *ηG2* = 0.02.

In AFAR, only the within-subjects ANOVA for posed data was significant, *F*(3, 18) = 3.75, *p* = 0.03, *ηG2* = 0.07. The multiple comparisons test did not reveal any significant differences, but the static model setting performed better than the dynamic one (Table 3). There were no other significant results, *F*s < 1.71, *p*s < 0.18, *ηG2*s < 0.02.

For FaceReader, none of the ANOVA results were significant, *F*s < 0.24, *p*s < 0.64, *ηG2*s < 0.00. As shown in Table 4, there was no significant difference between the “on” and “off” states, but there were small numerical differences (e.g., Aff-wild2 * AU25 = 0.71093 for off vs. 0.71097 for on).

## 4. Discussion

We used three databases with different contexts and compared the performance of the three AU detection systems: FaceReader, OpenFace, and AFAR. Comparison of the three systems shed light on the performance and confusion bias of each tool. Based on these results, we now discuss how to improve the algorithm and provide guidance for research on facial expressions. First, the results of one-sample *t*-tests comparing the AUC values showed that all AU detection systems were able to detect AUs correctly beyond the chance level, consistent with our hypothesis. Although it remains unclear with which database FaceReader was trained, at least two open-source systems (i.e., OpenFace and AFAR) were able to show better performance than chance with three databases that were not trained. Given that one database (Aff-wild2) is comprised of ecologically valid footage derived from YouTube [44] and the other one (GFT) comprised social conversation recorded in a natural environment, these results indicate the generalizability of the extant automatic AU systems.

As for comparisons of the three automated facial AU detection systems, the results indicated that FaceReader underperformed relative to the two open-source systems, and there were no significant differences between OpenFace and AFAR. From the viewpoint of AU detection accuracy, there is no significant difference between OpenFace and AFAR, but the number of AUs output by default is larger in OpenFace than in AFAR (18 vs. 12). Therefore, in terms of breadth of use, OpenFace may be superior to the other two systems. It should be noted that AFAR, which was trained using only one database, namely Expanded BP4D+, showed the same generalizability, although OpenFace trained using seven databases. Although fine-tuning is beyond the scope of the current study, it is expected that more fine-tuning of AFAR might close the gap with OpenFace, as [9] indicated.

Although we hypothesized that all machines would show better AU detection performance with the posed facial expression than with the spontaneous facial expressions, OpenFace and AFAR exhibited no significant difference in performance according to the type of database, which could be interpreted as supporting cross-corpora generalizability. However, FaceReader had the opposite result, with higher AUC scores for in-the-wild data pooled from YouTube than for the posed facial data. This result may suggest the direction that FaceReader should train. That is, to increase generalizability, it might be useful for FaceReader to train with facial data expressed in experimental and/or controlled situations, as the two open-software tools have.

The overall performance for AU12 was generally higher than that for other AUs, although AU6 detection accuracy for posed expressions (AUCs > 0.90 with OpenFace and AFAR) was higher than that of AU12. These AUs can be considered prototypical components of expressions of happiness [45,46], and the finding is consistent with the very high recognition rate of happy facial expressions previously reported for both automated emotion estimation systems and humans [16,47]. However, this study provided the first evidence that AU6 detection accuracy may be improved when targeting a posed database. Recently, this facial component has been discussed by many scholars [48,49,50,51], and the present result showing that automatic detection of AU6 is accurate when targeting facial expressions by models following instructions (i.e., posed expressions) may facilitate further research on facial expression or recognition.

On the other hand, detection of certain facial movements such as AU14 (dimpler: average AUC = 0.58) and 23 (lip tightener: average AUC = 0.56) seems difficult for automatic AU detection systems. These AUs appear to reflect facial tension rather than facial movement, suggesting that not only the movement vector, which can be detected by the landmark detection algorithm, but also the approach, which can capture undulations in facial movements, will be necessary. In fact, recent studies have suggested that human faces have distinctive undulations when expressing each facial component [52]. Therefore, further development of RGB analyses or other information to subtly differentiate these AUs is desired. For example, depth information on the spatial characteristics of facial objects in three dimensions (3D: see [53]) and rendering technology targeted to diffuse albedo that encodes blood flow [54] would be useful for accurately detecting wrinkles and tension in parallel.

As for the bias score, all systems over-predicted the co-occurrence of AU1 and AU2. This result seems reasonable, as the inner and outer eyebrows raiser commonly co-occur [55] under the training data situation, although in wild data and social conversation data, these facial components often occur independently. As shown in Appendix A, only AU1 occurred in Aff-wild2, and only AU2 occurred in GFT. This result may reveal the different patterns of utterance emphasis tied to context; i.e., AU1 might occur when speaking to unknown viewers (default YouTuber’s situation), and AU2 when interacting directly with a small number of people. To reduce the co-occurrence bias of AU1 and AU2, fine-tuning of GFT and Aff-wild2 might be useful. It may also be useful to train the models on facial stimuli by creating individual AUs with avatar generation software such as FACSGen [56,57].

Our results indicate the particular co-occurrence detection biases for each system. FaceReader was apt to detect AU7 associated with AU24. OpenFace and FaceReader tended to detect AU12 with AU25, whereas OpenFace and AFAR tend to detect AU14 at the same time as other AUs. These results provide information for users of these systems to determine in advance whether the AU to be detected is likely to be confused with another. Across bias patterns, AU14 induced considerable confusion in OpenFace and AFAR, which provided better performance than FaceReader, because the AU14 detection of automated AU detection systems was low. Future study will benefit from an ensemble system that can accurately detect wrinkles and tension, as noted above.

Through in-system comparisons, this study obtained two interesting findings: the calibration of FaceReader did not change the output results, and the static mode for AFAR and OpenFace was better than the dynamic mode for analyzing posed databases. The latter finding was particularly unexpected because the dynamic mode, which uses person-specific normalization, is considered especially suitable for dynamic facial databases. The results suggest that the appropriate processing method depends on the nature of the target facial expressions. Further studies will be necessary to fully understand how pre-processing should be applied according to the target facial expressions.

Table 5 shows the comprehensive study results. For the novice user, OpenFace may be the best choice regardless of the nature of the target expressions, since it has been trained on the most facial databases. However, as Table 2 and Table 3 show, using SIFT with AFAR is preferable when the user wishes to measure lip tension, e.g., AU23 (lip tightener) and 24 (lip pressor). The figures and tables in this paper could serve as references for future research. In the context of posed facial expressions, the static mode for OpenFace and AFAR will be useful.

Although we investigated the performance in identifying AUs over three dynamic facial databases, several limitations to this study must be noted. First, there were differences in the number of target frames among the three databases (in-the-wild = 170 K; posed = 12 K; conversation = 133 K). Therefore, the effect of posed expression on the bias score is relatively small. Spontaneous facial expressions often include few facial movements [2,58], and the overall duration of such expressions tended to be longer than that of posed facial expressions [59]. Therefore, differences in the number of frames between spontaneous and posed expressions are likely. Nevertheless, the lack of any significant difference in performance between posed and spontaneous databases for both OpenFace and AFAR supports the generalizability of these automated facial action detection systems. Future studies should include more posed facial databases to further investigate system performance.

Second, at present, facial databases with AU annotation are limited, and there are too few to ensure adequate learning and/or validation. The fact that the type and number of manually coded AUs depended heavily on databases may cause a difficult situation in the evaluation of automated AU detection systems. Several databases containing AU annotations exist, such as BP4D+ [39], EB+ [8], and FEAFA [60], but the number of AUs is restricted and/or unique AUs are included, so they are not unified. It is expected that more information can be obtained by performing analyses using existing databases. Furthermore, future study should develop and accumulate new facial databases that include AU annotations that can satisfy multiple needs.

Finally, the current study used FaceReader version 7. For comparison with state-of-the-art algorithms such as OpenFace and AFAR, FaceReader should be updated to version 8.1. Furthermore, many studies applied another commercial system (i.e., Affectiva: [61,62,63]) instead of FaceReader, but we could not include this system in the current study. For a comprehensive understanding of automated facial action detection systems, other commercial systems and facial expression analysis tools [64] should be included, as well as additional comparisons.

## 5. Conclusions

This study confirmed that the user-friendly automated facial AU detection systems can detect AUs more accurately than chance. In addition, a comparison of automatic AU detection systems showed that state-of-the art algorithms such as OpenFace and AFAR provided higher AUC scores than a commercial tool like FaceReader, but these two open-source tools also showed confusion regarding the tension-related facial component AU14, which was not seen by FaceReader. Finally, when analyzing the posed facial database using OpenFace or AFAR, the static rather than dynamic mode more accurately detected AUs. The study also provided clues that different AUs are expressed depending on the context. However, depending on the type of AU, the context, and so on, sufficiently accurate AU detection might still not be achieved. To further leverage the study of facial expressions and emotions, it is important to develop and improve tools for training/analyzing facial movements themselves.

## Figures and Tables

**Figure 1 sensors-21-04222-f001:**
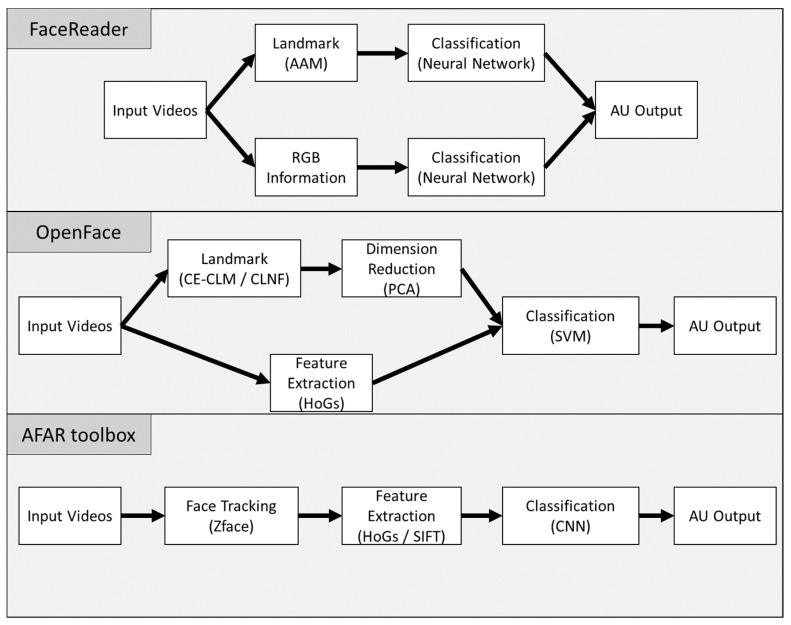
Pipelines of FaceReader, OpenFace, and the AFAR toolbox.

**Figure 2 sensors-21-04222-f002:**
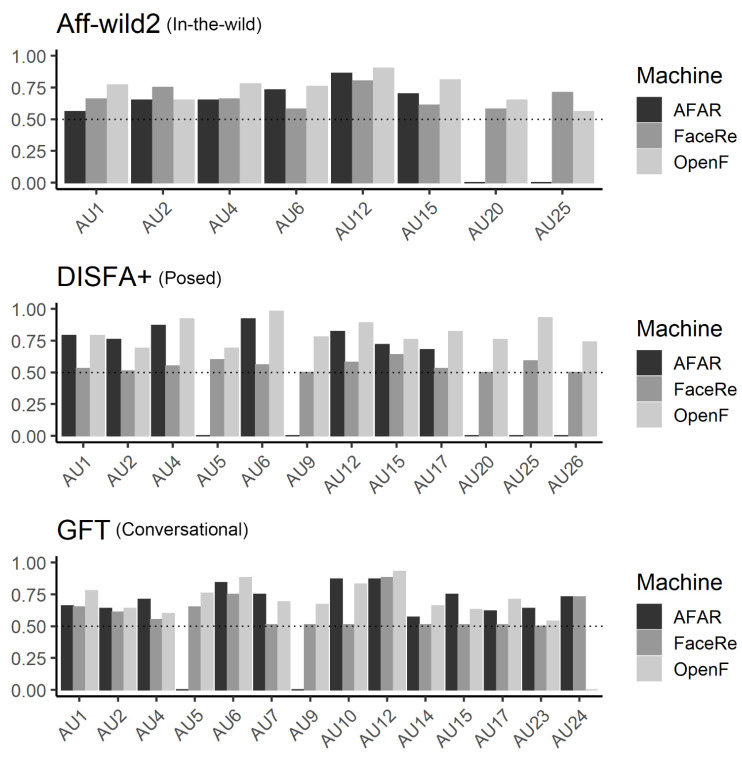
Average scores for each predicted AU, for all databases and machines. Dotted line represents random performance (i.e., AUC = 50). A zero value indicates an AU type that was not provided by the algorithm (e.g., AU20 in Aff-wild2*AFAR).

**Figure 3 sensors-21-04222-f003:**
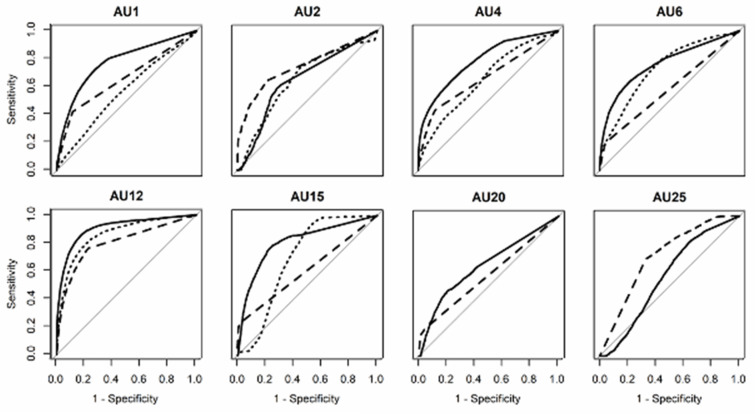
Receiver operating characteristic curves for each action unit in the Aff-wild2 (in-the-wild) database obtained by the three machines. Solid line represents OpenFace, dashed line represents FaceReader, and dotted line represents AFAR. The diagonal line indicates random performance).

**Figure 4 sensors-21-04222-f004:**
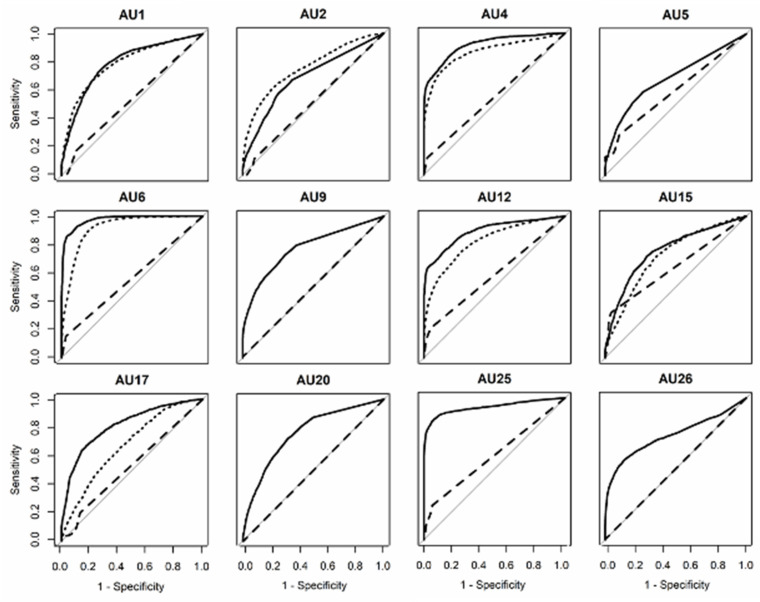
Receiver operating characteristic curves for each action unit in the DISFA+ (posed) database obtained by the three machines. Solid line represents OpenFace, dashed line represents FaceReader, and dotted line represents AFAR. The diagonal line indicates random performance.

**Figure 5 sensors-21-04222-f005:**
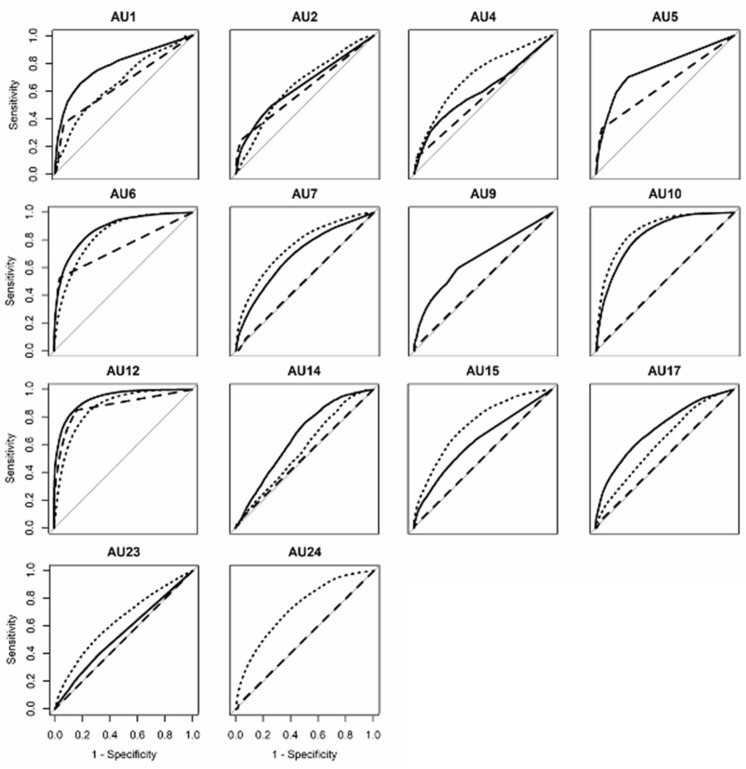
Receiver operating characteristic curves for each action unit in the GFT (conversational) database obtained by the three machines. Solid line represents OpenFace, dashed line represents FaceReader, and dotted line represents AFAR. The diagonal line indicates random performance.

**Figure 6 sensors-21-04222-f006:**
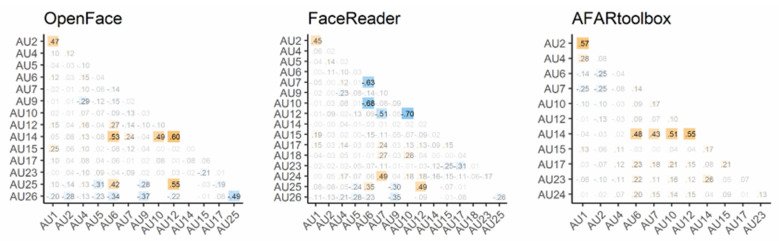
Bias scores of the three automated facial action unit detection systems. If this value is close to 0, the system obtains the same AU co-occurrence rate as the observational data. Positive values (orange squares) indicate biases that predict more AU co-occurrence than the actual data, and negative values (blue squares) indicate biases that fail to capture the co-occurrence relationships in actual facial expressions.

**Table 1 sensors-21-04222-t001:** AUs from the Facial Action Coding System [2]. The images were created using FACSGen.

AU	FACS Name	
AU1	Inner brow raiser	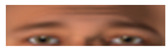
AU2	Outer brow raiser	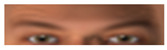
AU4	Brow lowerer	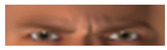
AU5	Upper lid raiser	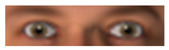
AU6	Cheek raiser	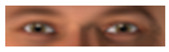
AU7	Lid tightener	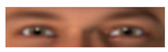
AU9	Nose wrinkler	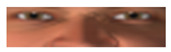
AU10	Upper lip raiser	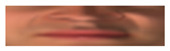
AU12	Lip corner puller	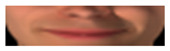
AU14	Dimpler	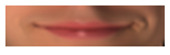
AU15	Lip corner depressor	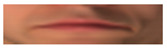
AU17	Chin raiser	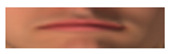
AU20	Lip stretcher	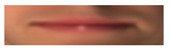
AU23	Lip tightener	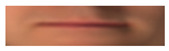
AU24	Lip pressor	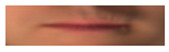
AU25	Lips part	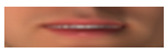
AU26	Jaw drop	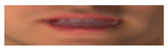

**Table 2 sensors-21-04222-t002:** Average AUC scores for each AU with all options activated in OpenFace: CE-CLM = convolutional experts constrained local model. CLNF = constrained local neural field model. The letter “s” denotes the analysis of static images. Bold font denotes the highest AUC values.

OpenFace/In-the-Wild					OpenFace/Posed				OpenFace/Conversation			
AU	CE-CLM	CE-CLM_s	CLNF	CLNF_s	CE-CLM	CE-CLM_s	CLNF	CLNF_s	CE-CLM	CE-CLM_s	CLNF	CLNF_s
AU1	**0.748**	0.690	0.696	0.632	0.791	**0.847**	0.786	0.841	**0.782**	0.653	0.763	0.647
AU2	0.648	0.698	0.632	**0.720**	0.689	**0.911**	0.702	0.912	0.637	**0.665**	0.625	0.643
AU4	0.778	**0.780**	0.743	0.743	0.917	0.917	**0.920**	**0.920**	**0.597**	0.596	0.578	0.578
AU5	-	-	-	-	0.688	**0.946**	0.680	0.937	**0.762**	0.741	0.761	0.715
AU6	0.769	**0.770**	0.759	0.759	0.976	0.976	**0.981**	**0.981**	**0.879**	0.878	0.870	0.870
AU7	-	-	-	-	-	-	-	-	**0.687**	**0.687**	0.685	0.685
AU9	-	-	-	-	0.775	0.984	0.767	**0.988**	0.671	**0.745**	0.663	0.743
AU10	-	-	-	-	-	-	-	-	**0.834**	0.833	0.824	0.824
AU12	0.874	**0.875**	0.831	0.831	0.894	0.894	**0.902**	**0.902**	**0.933**	0.932	0.927	0.927
AU14	-	-	-	-	-	-	-	-	0.654	**0.655**	0.627	0.627
AU15	**0.803**	0.723	0.600	0.688	0.756	**0.945**	0.757	0.943	0.627	0.690	0.618	**0.692**
AU17	-	-	-	-	0.817	0.867	0.853	**0.869**	**0.712**	0.652	0.710	0.641
AU20	**0.622**	0.534	0.582	0.525	**0.760**	0.718	0.742	0.668	-	-	-	-
AU23	-	-	-	-	-	-	-	-	0.544	**0.563**	0.547	0.558
AU25	0.544	**0.603**	0.521	0.541	0.935	**0.959**	0.936	0.957	-	-	-	-
AU26	-	-	-	-	0.736	0.722	**0.759**	0.757	-	-	-	-
Mean	**0.723**	0.709	0.670	0.680	0.811	**0.891**	0.816	0.890	**0.717**	0.715	0.708	0.704

**Table 3 sensors-21-04222-t003:** Average AUC scores for each AU with all options activated in AFARtoolbox: HOG = histograms of oriented gradient. SIFT = scale-invariant feature transform. The letter “s” denotes the analysis of static images. Bold font denotes the highest AUC values.

AFAR/In-the-Wild					AFAR/Posed				AFAR/Conversation			
AU	HOG	HOG_s	SIFT	SIFT_s	HOG	HOG_s	SIFT	SIFT_s	HOG	HOG_s	SIFT	SIFT_s
AU1	**0.569**	0.501	**0.569**	0.501	0.789	0.780	**0.790**	0.780	**0.780**	0.640	0.778	0.636
AU2	0.521	**0.524**	0.523	0.523	0.756	0.833	0.756	**0.834**	0.713	0.616	**0.717**	0.608
AU4	**0.677**	0.625	0.676	0.624	**0.870**	0.828	0.869	0.828	0.707	0.703	0.702	**0.712**
AU6	0.663	**0.666**	0.662	**0.666**	0.919	**0.949**	0.919	**0.949**	0.829	0.810	0.841	**0.846**
AU7	-	-	-	-	-	-	-	-	0.744	0.735	0.745	**0.757**
AU10	-	-	-	-	-	-	-	-	0.824	0.843	0.846	**0.873**
AU12	0.706	**0.743**	0.706	**0.743**	0.823	**0.851**	0.823	0.850	0.832	0.838	0.857	**0.863**
AU14	-	-	-	-	-	-	-	-	0.581	0.561	**0.612**	0.605
AU15	**0.761**	0.722	0.760	0.722	0.720	0.801	0.720	**0.802**	0.664	0.726	0.664	**0.732**
AU17	-	-	-	-	0.675	**0.776**	0.676	**0.776**	0.641	0.612	0.633	**0.648**
AU23	-	-	-	-	-	-	-	-	0.646	0.620	**0.662**	0.643
AU24	-	-	-	-	-	-	-	-	0.696	0.707	0.709	**0.764**
Mean	**0.649**	0.630	**0.649**	0.630	0.793	**0.831**	0.793	**0.831**	0.721	0.701	**0.730**	0.724

**Table 4 sensors-21-04222-t004:** Average AUC scores for each AU in FaceReader: Default indicates that there was no calibration.

	FaceReader/In-the-Wild		FaceReader/Posed		FaceReader/Conversation	
AU	Default	Calibration	Default	Calibration	Default	Calibration
AU1	0.656	0.656	0.533	0.533	0.649	0.649
AU2	0.746	0.746	0.513	0.513	0.606	0.606
AU4	0.663	0.663	0.549	0.549	0.551	0.551
AU5	-	-	0.596	0.596	0.645	0.645
AU6	0.579	0.579	0.561	0.561	0.747	0.747
AU7	-	-	-	-	0.507	0.507
AU9	-	-	0.504	0.504	0.510	0.510
AU10	-	-	-	-	0.506	0.506
AU12	0.800	0.800	0.583	0.583	0.877	0.877
AU14	-	-	-	-	0.513	0.513
AU15	0.607	0.607	0.639	0.639	0.505	0.505
AU17	-	-	0.530	0.530	0.508	0.508
AU20	0.579	0.579	0.498	0.498	-	-
AU23	-	-	-	-	0.500	0.500
AU24	-	-	-	-	0.506	0.506
AU25	0.711	0.711	0.592	0.592	-	-
AU26	-	-	0.501	0.501	-	-
Mean	0.667	0.667	0.550	0.550	0.581	0.581

**Table 5 sensors-21-04222-t005:** Overview of the results of the current study. A, average AUC score > 0.80; B, average AUC score = 0.71–0.80; C, average AUC score = 0.61–0.70; D, average AUC score < 0.60.

System	In-the-Wild	Posed	Conversational	Remarks
FaceReader	C	D	D	This system also can estimate emotional states (emotion category, valence and arousal).
OpenFace	B	A	B	For analyzing posed facial data, static mode is better than dynamic mode.
AFAR toolbox	C	B	B	For analyzing posed facial data, static mode is better than dynamic mode.

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
