# Peer review of "Assessing Automated Facial Action Unit Detection Systems for Analyzing Cross-Domain Facial Expression Databases"

_sensors, 2021, doi:10.3390/s21124222_

Round 1

Reviewer 1 Report

Very good paper. A few slight comments:

  • I found myself flipping back and forth to recall which DB was posed/conversational/etc. Surely this is the most important aspect of the DBs and I would label Figures 1 and 2 to show this information more conveniently.
  • I would also make the small images in Table 1 much bigger to increase clarity (go to two columns?)
  • There is also a real disparity between data size in the DISFA and the other databases; any discussion of the impact of this (if any)?

Author Response

We are grateful for the extremely helpful comments of the reviewers. We have addressed all of the issues highlighted, and believe that the manuscript has been improved considerably as a result of these changes.

Point 1

Very good paper. A few slight comments:

I found myself flipping back and forth to recall which DB was posed/conversational/etc. Surely this is the most important aspect of the DBs and I would label Figures 1 and 2 to show this information more conveniently.

Response

Following the reviewer’s comment, we added labels to revised Figures 3-5.

Point 2

I would also make the small images in Table 1 much bigger to increase clarity (go to two columns?)

Response

Thank you for the comment. As suggested, we increased the size of images in Table 1.

Point 3

There is also a real disparity between data size in the DISFA and the other databases; any discussion of the impact of this (if any)?

Response

This is an important point. As the reviewer noted, the difference in number of frames among the three databases is a limitation of this study, especially with respect to the determination of bias. We have clearly explained this limitation in the Discussion section of the revised manuscript (Page 8, Lines 375–384).

Reviewer 2 Report

The manuscript addresses a viable topic in Affective Computing and has promising prospects. However, authors need to address certain concerns.

  1. Firstly, there is a mention of two commercial systems for automatic AU detection systems, Affectiva and FaceReader. Since the authors have mentioned these systems, the study for the Affectiva system should also be included for the same experiments for a fair comparison study.
  2. The work is good but lacks the significance of the study that has been done. A good way to address this concern can be to include data mentioning the use cases of this study, as to how a reader can benefit after reading this study. Something like the scenarios in which an Affective system would prove more beneficial than others. Also, the discussion section can include a table to present the information there in a more readable manner.
  3. There is inconsistency in certain terms being used in the manuscript. It should be the same to avoid any ambiguity. For e.g., line 61 says spontaneous expressions while line 62 refers to them as natural expressions.
  4. It is not clear in the beginning what is meant by facial displays. This creates confusion and it should be mentioned what is actually being referred to here.
  5. The acronyms have not been handled properly. For e.g., the AUC acronym appears in line 16 before specifying what it means. Similarly, ROC has been used before specifying its full form. Moreover, at some places, the full form is being used and at some places, the acronym is being used. Line 62 refers to facial expressions and line 94 refers to them as facial reactions. These should be addressed and should be consistent. 
  6. There are lots of grammatical mistakes and the paper should be improved in terms of grammar. For e.g. line 39 needs modification.
  7. The references are missing at some places, for e.g., line 83 should be cited properly. Line 91 mentions previous work but has not been cited.
  8. Wrong reference to figures in line 210
  9. The references are not formatted correctly. Some years are in bold and some in normal. Consistency needs to be maintained.

Author Response

We are grateful for the extremely helpful comments of the reviewers. We have addressed all of the issues highlighted, and believe that the manuscript has been improved considerably as a result of these changes.

Point 1

The manuscript addresses a viable topic in Affective Computing and has promising prospects. However, authors need to address certain concerns.

Firstly, there is a mention of two commercial systems for automatic AU detection systems, Affectiva and FaceReader. Since the authors have mentioned these systems, the study for the Affectiva system should also be included for the same experiments for a fair comparison study.

Response

This is an important point. Although we acknowledge that the Affectiva system should ideally also have been included, this was not possible due to budget constraints. We state this clearly in the Discussion section (Page 8, Lines 396–400).  

Point 2

The work is good but lacks the significance of the study that has been done. A good way to address this concern can be to include data mentioning the use cases of this study, as to how a reader can benefit after reading this study. Something like the scenarios in which an Affective system would prove more beneficial than others. Also, the discussion section can include a table to present the information there in a more readable manner.

Response

Thank you for your comments. To address the reviewer’s concern, we compare the systems in the Results section (Pages 6, Lines 260–267).

In addition, we added text to the Discussion section to address the other points raised (Pages 8, Lines 367–373) and made a new overview table in Table 4.

Point 3

There is inconsistency in certain terms being used in the manuscript. It should be the same to avoid any ambiguity. For e.g., line 61 says spontaneous expressions while line 62 refers to them as natural expressions.

It is not clear in the beginning what is meant by facial displays. This creates confusion and it should be mentioned what is actually being referred to here.

The acronyms have not been handled properly. For e.g., the AUC acronym appears in line 16 before specifying what it means. Similarly, ROC has been used before specifying its full form. Moreover, at some places, the full form is being used and at some places, the acronym is being used. Line 62 refers to facial expressions and line 94 refers to them as facial reactions. These should be addressed and should be consistent.

Response

Thank you for this comment, according to which we have carefully checked manuscript. For example, we now use the phrase “facial expression” instead of “facial display” and “facial reaction”. Additionally, we have defined “facial components” in terms of action units, and “facial movements” as the single or multiple facial components. As suggested, we have also modified the acronyms and keywords.

Point 4

There are lots of grammatical mistakes and the paper should be improved in terms of grammar. For e.g. line 39 needs modification.

Response

The  English in the revised document has been checked by at least two professional editors, both native speakers of English. For a certificate, please see https://osf.io/84rn2/

Point 5

The references are missing at some places, for e.g., line 83 should be cited properly. Line 91 mentions previous work but has not been cited.

Response

We have added the appropriate references to the Introduction section (Page 2, Lines 82-83 and 92).

Point 6

Wrong reference to figures in line 210

Response

We have modified the Figure references in the Result section (Page 5, Lines 228-229).

Point 7

The references are not formatted correctly. Some years are in bold and some in normal. Consistency needs to be maintained.

Response

We have modified the references according to this comment.

Reviewer 3 Report

This paper compares the performance of three popular AU detection tools. The author addresses three major concerns of these tools, including: generalization ability across various domains; systematic accuracy comparison; and detection bias. The author does several experiments regarding the above aspects and give valuable discussion on how the detection method could be improved.

However, there're some aspects that could be improved. First, there's few analysis on the algorithm used by these tools. It seems the three tools are used for general facial analysis. Thus, AU detection is only one component and could use different algorithm in different software versions. Analysis and comparison on the specific algorithm would bring more insight to the paper. Second, this paper could be useful to both readers that want to learn more about the detection tools, and the AU detection algorithm community. I suggest the author add a specific section on how algorithm could be improved on each AU. Thirdly, the writing could be improved. Some sentences are not clear. For example: Moreover, two state-of-the art algorithms provided higher AUC scores than a commercial tool. (line 15). It's not clear which two algorithms are better than which tool, and are these algorithms discussed in the paper?

Author Response

We are grateful for the extremely helpful comments of the reviewers. We have addressed all of the issues highlighted, and believe that the manuscript has been improved considerably as a result of these changes.

Point 1

However, there're some aspects that could be improved. First, there's few analysis on the algorithm used by these tools. It seems the three tools are used for general facial analysis. Thus, AU detection is only one component and could use different algorithm in different software versions. Analysis and comparison on the specific algorithm would bring more insight to the paper.

Response

As suggested, further analysis of the algorithm was performed, and the features of the tools were compared (see the Results section: Page 6, Lines 260–282).

Figure 1 now illustrates the pipeline of each system (Page 3, Lines 100-106 and 136–137).

Point 2

Second, this paper could be useful to both readers that want to learn more about the detection tools, and the AU detection algorithm community. I suggest the author add a specific section on how algorithm could be improved on each AU.

Response

We have described three ways to improve AU detection. The first is to choose the tools and/or algorithm appropriately, according to the use case (Page 8, Lines 367–373). For example, among the three tools, AFAR is best suited to detect mouth tension (AU23,24). At the user level, this information is important.

The second way to improve AU detection is to increase the number of features that can be extracted and used for AU detection (Page 7, Lines 342–347). We focused on capturing undulations in facial movements using 3D information or diffuse albedo, which can provide useful information for developers of AU detection.

The third way is to use software to generate facial avatars, to avoid the co-occurrence bias seen in the current study (Page 7, Lines 346–349).

Point 3

Thirdly, the writing could be improved. Some sentences are not clear. For example: Moreover, two state-of-the art algorithms provided higher AUC scores than a commercial tool. (line 15).

Response

Following the reviewer’s comment, we have revised the text in question for clarity (Page 1, Lines 15-16).

In addition, the English has been checked by at least two professional editors, both native speakers of English. For a certificate, please see https://osf.io/84rn2/.

Point 4

It's not clear which two algorithms are better than which tool, and are these algorithms discussed in the paper?

Response

We have clarified which algorithm was superior in the revised Discussion (Pages 8, Lines 368-373) and included a new Table 4.

Reviewer 4 Report

This manuscript aims to compare three AU detection systems. The manuscript utilized mainly the AUC and a co-occurrence score for performance comparison.

The contribution of this study is not significant enough to be published in a journal yet. My views to improve the content of this manuscript are as follows:

  1. There is no novelty presented in the manuscript.
  2. Although the manuscript describes briefly each tool, there is no systematic approach provided for selection of three AU detection systems. The reader would be interested in comparison of detection tools that utilize different processing pipelines from the preprocessing to the end of detection. Comparison of the algorithms which are implemented in each detection tool should be particularly in focus. In addition, since the type and the size of the training data sets are different, as indicated in the manuscript, the performance of a tool would outperform the others as expected; therefore, the benefit of comparing tools in such a setup is not clear for the scientific world.
  3. The comparison approach is straightforward. The co-occurrence measure needs to be described clearly with an adequate formulation. The readers would be interested in comprehensive comparison beyond just performance. It would be very interesting to compare pipelines/algorithms/tools after grouping videos/images by facial expressions and/or face types.

Author Response

We are grateful for the extremely helpful comments of the reviewers. We have addressed all of the issues highlighted, and believe that the manuscript has been improved considerably as a result of these changes.

Point 1

The contribution of this study is not significant enough to be published in a journal yet. My views to improve the content of this manuscript are as follows:

  1. There is no novelty presented in the manuscript.

Response

Thank you for your comments. We have clarified the novel aspects of our study in the Discussion section of the revised manuscript (Page 6, Lines 285-288).

Point 2

  1. Although the manuscript describes briefly each tool, there is no systematic approach provided for selection of three AU detection systems. The reader would be interested in comparison of detection tools that utilize different processing pipelines from the preprocessing to the end of detection. Comparison of the algorithms which are implemented in each detection tool should be particularly in focus. In addition, since the type and the size of the training data sets are different, as indicated in the manuscript, the performance of a tool would outperform the others as expected; therefore, the benefit of comparing tools in such a setup is not clear for the scientific world.

Response

This is an important point. Thus, we have compared the features available among the tools (see the Results section: Page 6, Lines 260-282).

Also, Figure 1 now describes the pipelines of each system (Page 3, Lines 100-104 and 136-137), and the system are explained in more detail in the Method section (Page 4, Lines 148-151 and 162-165).

When analyzing the posed facial database using OpenFace or AFAR, static mode detected AUs more accurately than dynamic mode. This important point is discussed in a new paragraph in the Discussion section (Page 8, Lines 359-366), and in the Abstract (Page 1, Lines 18-19) and Conclusions (Page 9, Lines 407-409). Thank you for your insightful comments.

Point 3

  1. The comparison approach is straightforward. The co-occurrence measure needs to be described clearly with an adequate formulation. The readers would be interested in comprehensive comparison beyond just performance. It would be very interesting to compare pipelines/algorithms/tools after grouping videos/images by facial expressions and/or face types.

Response

We have added a formulation for the co-occurrence measure to the Methods section (Page 4, Lines 184-186).

Moreover, for comprehensive comparison, we created a new Table 4 and summarize our results therein (Pages 8, Lines 367-373), to provide guidance on the selection of automated facial detection.

We agree that comparing pipelines/algorithms/tools after grouping videos/images by facial expressions is an interesting idea. Thus, we further analyzed the posed database (DISFA+), by dividing it into six emotional categories. We have added the results of this comparison to the Supplemental Material, and a csv file that includes all of the AU data (https://osf.io/7rguf/?show=view).

Round 2

Reviewer 3 Report

The authors have enhanced the article with more discussion and results.

Author Response

Point 1

The authors have enhanced the article with more discussion and results.

Response

We are honored to hear that. Thank you for your kind feedback and helpful comments.

Reviewer 4 Report

This manuscript aims to compare three AU detection systems. The manuscript utilized mainly the AUC and a co-occurrence score for performance comparison.

The contribution of this study is not significant enough to be published in a journal yet. My views to improve the content of this manuscript are as follows:

  1. The formula for the co-occurrence is not written in a proper way. Authors should be well aware of the basic technical writing style. As an example,

C = … / ….      (#)

where .. denotes …

  1. The denominator takes the square root of “…AUx* - meanAUx* ..” but not that of “AUy*”. Also, the symbol for “sum” applies to “AUx*” only but not to “AUy*”. The formula must be corrected.
  2. Where does “six emotional categories” presented in the manuscript?

“We agree that comparing pipelines/algorithms/tools after grouping videos/images by facial expressions is an interesting idea. Thus, we further analyzed the posed database (DISFA+), by dividing it into six emotional categories.”

4. The sentence on the page 4 at the line 186 should be revised. “On the other hand, if the absolute value is close to 185 1, the result can be considered the difference between the actual data and the machine.”

Author Response

We are grateful for the extremely helpful comments of the reviewers. We have addressed all of the issues highlighted, and believe that the manuscript has been improved considerably as a result of these changes.

Point 1

The formula for the co-occurrence is not written in a proper way. Authors should be well aware of the basic technical writing style. As an example,

C = … / ….      (#)

where .. denotes …

The denominator takes the square root of “…AUx* - meanAUx* ..” but not that of “AUy*”. Also, the symbol for “sum” applies to “AUx*” only but not to “AUy*”. The formula must be corrected.

Response

Thank you very much for your helpful comments. Following the reviewer’s suggestion, we modified the relevant styles of the formula (Page 4, Lines 182-185). Please see the followings:

Page 4, Lines 182-185

------------------------------------------------

(1: Formula is included in word file)

where  is the xth AU that the system has predicted or actually observed, and  is the yth AU that the system has predicted or actually observed.

------------------------------------------------

Point 2

Where does “six emotional categories” presented in the manuscript?

“We agree that comparing pipelines/algorithms/tools after grouping videos/images by facial expressions is an interesting idea. Thus, we further analyzed the posed database (DISFA+), by dividing it into six emotional categories.”

Response

Thank you for your comments. We have added an explanation of these results to the Results section (Page 5, Lines 229-234).

Page 5, Lines 229-234

------------------------------------------------

Furthermore, to provide information for affective computing, we compared the AUC values obtained by tools after grouping videos by emotions in the DISFA+ posed facial database. Since it may be beyond the scope of this paper, we have not shown these results. More detailed results have been made available online as supplementary material (https://osf.io/vuerc/?view_only=8e9077dbeaec424bad5dde5f07555677).

------------------------------------------------

Point 3

The sentence on the page 4 at the line 186 should be revised. “On the other hand, if the absolute value is close to 185 1, the result can be considered the difference between the actual data and the machine.”

Response

By following the reviewer’s comment, we revised the sentence (Page 4, Lines 187-188).

Page 4, Lines 187-188

------------------------------------------------

On the other hand, if the absolute value is close to 1, this indicates that the machine was biased or overlooked AU co-occurrence.

------------------------------------------------
